# Ultrasensitive detection of miRNA with an antimonene-based surface plasmon resonance sensor

Tianyu Xue[1], Weiyuan Liang[1], Yawen Li[2], Yuanhui Sun[2], Yuanjiang Xiang[1], Yupeng Zhang[1], Zhigao Dai[3,4], Yanhong Duo[1], Leiming Wu[1], Kun Qi[1], Bannur Nanjunda Shivananju[1], Lijun Zhang [2], Xiaoqiang Cui[2], Han Zhang [1] & Qiaoliang Bao [3]

MicroRNA exhibits differential expression levels in cancer and can affect cellular transformation, carcinogenesis and metastasis. Although fluorescence techniques using dye molecule labels have been studied, label-free molecular-level quantification of miRNA is extremely challenging. We developed a surface plasmon resonance sensor based on two-dimensional nanomaterial of antimonene for the specific label-free detection of clinically relevant biomarkers such as miRNA-21 and miRNA-155. First-principles energetic calculations reveal that antimonene has substantially stronger interaction with ssDNA than the graphene that has been previously used in DNA molecule sensing, due to thanking for more delocalized $5s/5p$ orbitals in antimonene. The detection limit can reach 10 aM, which is 2.3–10,000 times higher than those of existing miRNA sensors. The combination of not-attempted-before exotic sensing material and SPR architecture represents an approach to unlocking the ultrasensitive detection of miRNA and DNA and provides a promising avenue for the early diagnosis, staging, and monitoring of cancer.

[1] Key Laboratory of Optoelectronic Devices and Systems of Ministry of Education and Guangdong Province, College of Electronic Science and Technology and College of Optoelectronics Engineering, Shenzhen University, 518060 Shenzhen, People's Republic of China. [2] School of Materials Science and Engineering and Key Laboratory of Automobile Materials of MOE, Jilin University, 130012 Changchun, Jilin, People's Republic of China. [3] Department of Materials Science and Engineering, ARC Centre of Excellence in Future Low-Energy Electronics Technologies (FLEET), Monash University, Clayton, VIC 3800, Australia. [4] School of Printing and Packaging and School of Physics and Technology, Wuhan University, 299 Bayi Road, Wuchang District, 430072 Wuhan, Hubei Province, People's Republic of China. Correspondence and requests for materials should be addressed to L.Z. (email: lijun_zhang@jlu.edu.cn) or to X.C. (email: xqcui@jlu.edu.cn) or to H.Z. (email: hzhang@szu.edu.cn) or to Q.B. (email: qiaoliang.bao@monash.edu)

Biomarkers have potential in the prediction, diagnosis and monitoring of diseases[1]. In particular, there is a need to discover biomarkers that can be used to detect diseases for which an early diagnosis is crucial or diagnosis is currently difficult[2]. MicroRNA (miRNA), which constitutes a class of short RNA, is emerging as ideal candidates as noninvasive biomarkers for applications in toxicology, diagnosis, and monitoring treatment responses or adverse events[3,4]. The aberrant expression of miRNA has been found in all types of tumours, including pancreatic cancer, lung cancer, prostate cancer, colorectal cancer, triple-negative breast cancer and osteosarcoma[4]. The detection of tumour-specific circulating miRNA at an ultrahigh sensitivity is of utmost significance for the early diagnosis and monitoring of cancer[5]. Unfortunately, miRNA detection remains challenging because miRNA are present at low levels and comprise ∼0.01% of the total RNA mass in a given sample. Therefore, the development of new approaches or sensing media for miRNA detection at the molecular level is urgently needed for clinical disease diagnosis.

Traditionally, the use of miRNA detection techniques, such as quantitative real-time PCR (qRT–PCR)[6], northern blotting[7] and microarray-based hybridization[8], is limited in early diagnosis in clinical practice due to the difficulty in amplification, the high cost, complex operations and low sensitivity. Due to its many advantages, such as non-destructive label-free detection, high reproducibility and low cost, the surface plasmon resonance (SPR) technique has proven to be versatile in investigations of molecular interactions by assessing the refractive index change on a chip surface[9–11]. Nevertheless, using the traditional SPR technique to detect biomolecules at very low concentrations remains challenging due to the limited quantity of immobilized probe DNA and miRNA on the chip surface (normally a thin gold film)[12–17]. Therefore, there is an urgent need to identify advanced material with large adsorption energy and work function increment to improve the performance of the SPR biosensor. Recently, numerous emerging two-dimensional (2D) nanomaterials have been tested for DNA molecule sensing, including graphene[18–20], transition-metal dichalcogenides (TMDs)[21,22], topological insulators[23,24], black phosphorus[25,26] and MXenes[27]. However, most nanomaterials are subject to certain limitations due to weak interactions with biomolecules or poor chemical stability. Identifying a new 2D material with a stronger molecular-level interaction with biomarkers is critical.

Antimonene has been described as a 2D material that can be exfoliated from bulk antimony (Sb) and has quickly attracted the attention of the scientific community because its physicochemical properties are superior to those of typical 2D materials (e.g., graphene, MoS$_2$, and black phosphorus)[28,29]. Similar to graphene materials, antimonene has an $sp^2$-bonded honeycomb lattice, but antimonene exhibits strong spin–orbit coupling, tremendous stability and hydrophilicity that is significantly better than that of graphene[29,30]. Antimonene nanosheets and quantum dots have already been used in nonlinear optics[31], photothermal therapy (PTT)[32], thermophotovoltaic (TPV) cells[33], and field effect transistors (FET)[34]. Although the photoelectrical properties of antimonene nanomaterials have been studied, the interaction between DNA and antimonene and its application in optical sensing remain elusive.

Here, we firstly explored via first-principles density functional theory (DFT) calculations the chemical interactions of single-stranded DNA (ssDNA) and double-stranded DNA (dsDNA) with antimonene, and find that antimonene has much better sensitivity than graphene previously used in DNA molecule sensing. Motivated by this theoretical finding, we developed a SPR sensor by using antimonene materials and performed trace attomolar-level quantification of miRNA molecules. This method

reached an extremely low limit of detection (LOD), surpassing that of existing sensing methods. In addition, the sensor can distinguish miRNA that differ by one nucleobase mutation. Because of the extremely large adsorption energy between ssDNA and antimonene, we can envision an ultrasensitive RNA and DNA sensor device for early cancer diagnosis. This proposed methodology based on antimonene materials for nucleic acid detection holds intriguing potential for the development of multiplexed lab-on-chip platforms, which can be further applied for clinical purposes.

## Results

**First-principles calculations**. DFT-based energetic calculations including dispersive Van der Waals forces were performed to investigate the differential interactions of ssDNA and dsDNA with antimonene, as summarized in Fig. 1. The changes in work function after ssDNA/dsDNA absorption are shown in Table 1. The interactions with graphene were studied for direct comparison. The adsorption energies ($E_{ad}$) of the bases on antimonene are higher than those of the base-pairs on antimonene, indicating the stronger interaction between the nucleobases and antimonene. The work function ($\Delta W$) shows substantial increase after DNA absorption. These behaviors are consistent with those of current and previous calculations of graphene-based systems[35]. Distinctly and importantly, we found that by comparison with graphene, antimonene exhibits the much stronger interaction with ssDNA, as indicated by the higher adsorption energies (for the ssDNA absorption case, Fig. 1g) as well as the about 1.5 times larger work function increment (Table 1). This is further supported by the charge density difference map between antimonene/graphene with DNA absorption and noninteracting counterparts (Fig. 1h), where the stronger charge transfer and electronic orbital hybridization occurs in the antimonene case (upper panel). Further calculations of adsorption energies and work functions for the nucleobases on top of antimonene/graphene with varied adsorption orientations indicate that the bases adsorption orientations have negligible effect on the above results obtained (see Supplementary Fig. 1). The underlying mechanism might be related to the more delocalized 5s/5p orbitals or the buckling honeycomb lattice of antimonene. These results indicate that antimonene is more sensitive to ssDNA than graphene in terms of sensing ability.

**Methodology of an antimonene-based miRNA sensor**. The strategy adopted for the highly sensitive detection of miRNA hybridization events on antimonene-modified SPR chips is depicted in Fig. 2. The SPR signal is sensitive to changes in the refractive index of the analyte. First, AuNRs are employed to connect with ssDNA to amplify the SPR signal. Then, AuNR-ssDNA complex is adsorbed onto the antimonene nanosheet due to the strong interaction between ssDNA and antimonene. Following the addition of complementary miRNA, the hybridized targets are easily desorbed from the antimonene interface since double-stranded DNA has a low affinity to antimonene. The amount of miRNA can be typically determined based on the negative shift of the SPR signal. The investigation of AuNR-ssDNA-modified antimonene interfaces is beneficial for detecting miRNA hybridization events.

**Preparation of the few-layer antimonene**. Liquid-phase sonication is an effective method for preparing few-layer antimonene by breaking weak van der Waals forces. We obtained few-layer antimonene nanosheets using sonication liquid exfoliation as shown in Fig. 3a. The antimonene nanosheets consist of β-phase antimony based on the hexagonal coordinate system. The

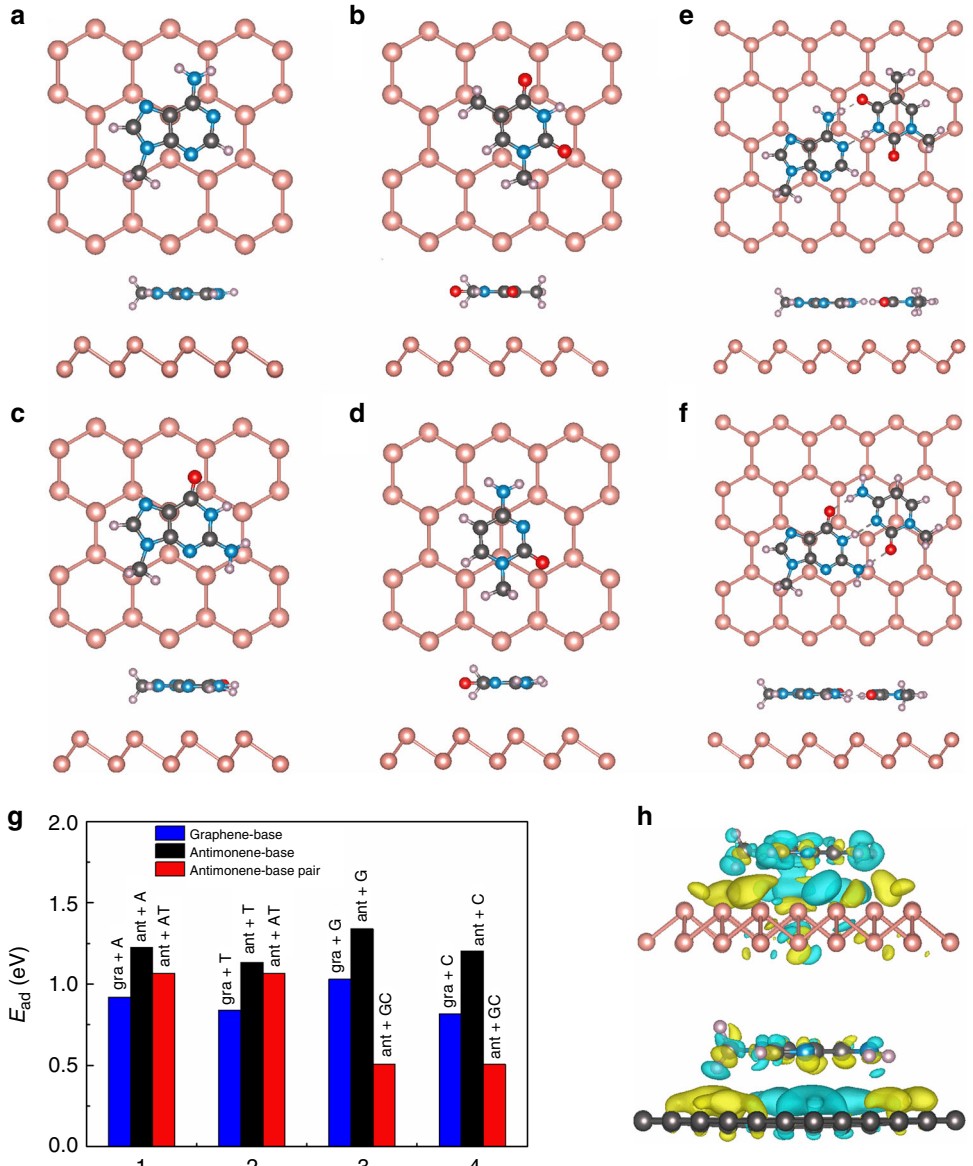

**Fig. 1** Geometry and energies of adsorption systems. **a** Top and side views of the optimized structure of A nucleobases on antimonene. **b** Top and side views of the optimized structure of T nucleobases. **c** Top and side views of the optimized structure of G nucleobases. **d** Top and side views of the optimized structure of C nucleobases. **e** Top and side views of the optimized structure of A–T base-pairs on antimonene. **f** Top and side views of the optimized structure of G–C base-pairs on antimonene. **g** Adsorption energies of adsorbed nucleobases and base-pairs on antimonene and graphene. Blue bars respect to graphene+A, black bars respect to antimonene+A, red bars respect antimonene+A–T. **h** Side views of the charge density difference of the nucleobases on antimonene and graphene

chemical composition and morphology of the prepared samples were systematically investigated. As shown in Fig. 3b, the Faraday–Tyndall effect, which is known as light scattering by particles in a fine suspension, was clearly observed, indicating the existence of antimonene nanosheets in the solution. Figure 3c shows a transmission electron microscope (TEM) image of the few-layer antimonene flake, in which very thin 2D nanosheets are resolved. The high-resolution TEM (HRTEM) image shows that the lattice is pure without any defects. The Fast Fourier Transform (FFT) image shows that the single-crystalline antimonene is a cubic system (Fig. 3d). The atomic force microscope (AFM) topography of typical antimonene nanosheets is shown in Fig. 3e. The overall lateral dimensions of the nanosheets are greater than 300 nm, and the thinnest piece is ~3 nm thick.

The crystal structure of antimonene was confirmed by XRD spectrum[36] as depicted in Fig. 3f. The diffraction peaks of

antimonene were identical to the spectrum of $\beta$-Sb precursor (JCPDS No. 35-0732). To further investigate the crystal structure and quality of the antimonene nanosheets, the Raman spectra of typical few-layer antimonene and bulk antimony were measured (Fig. 3g). Two characteristic Raman peaks, i.e., $E_g$ at 117 cm$^{-1}$ and $A_{1g}$ at 153 cm$^{-1}$, were observed in the Raman spectra of the few-layer antimonene. The degenerate modes of $E_g$ symmetry, which correspond to the in-plane transversal and longitudinal vibrations of the sublayers in opposite directions, cause the experimentally observed Raman peak at 117 cm$^{-1}$. The peak at 153 cm$^{-1}$ is caused by the third mode opposite-in-phase out-of-plane vibrations of the sublayers of the $A_{1g}$ symmetry[37]. Compared to the bulk material, in the few-layer system, a strong contraction of the in-plane lattice constant occurred as the film thickness decreased. Thus, the bulk $A_{1g}$ mode blue shifted from 150 cm$^{-1}$ to 153 cm$^{-1}$, and the film thickness decreased. The

**Table 1 Calculated vertical distance (Å) and work function change (ΔW) of antimonene and graphene with nucleobases and base-pairs**

|  |  | A | T | G | C | A–T | G–C |
|---|---|---|---|---|---|---|---|
| Antimonene | Distance (Å) | 3.5 | 3.65 | 3.48 | 3.5 |  |  |
|  | ΔW (eV) | 0.093 | 0.12 | 0.13 | 0.096 | 0.104 | 0.071 |
| Graphene | ΔW (eV) | 0.075 | 0.083 | 0.086 | 0.045 | 0.064 | 0.055 |

The calculated work functions of isolated antimonene and graphene are 4.389 and 4.208 eV, respectively

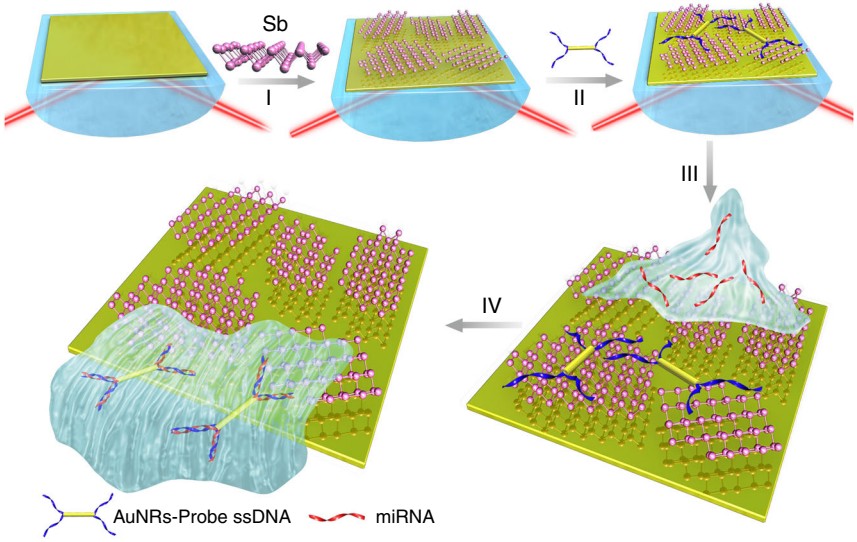

**Fig. 2** Fabrication of a miRNA sensor integrated with antimonene nanomaterials. Schematic illustration of the strategy employed to detect antimonene-miRNA hybridization events. I The antimonene nanosheets was assembled on the surface of Au film. II AuNR-ssDNAs were adsorbed on the antimonene nanosheets. III miRNA solution with different concentrations flowed through the surface of antimonene, and paired up to form a double-strand with complementary AuNR-ssDNA. IV The interaction between miRNA with AuNR-ssDNA results in release of the AuNR-ssDNA from the antimonene nanosheets. The reduction in the molecular of the AuNR-ssDNA on the SPR surface makes for a significant decrease of the SPR angle

chemical compositions of the prepared antimonene were confirmed by X-ray photoelectron spectroscopy (XPS) as shown in Fig. 3h. The two characteristic peaks at 528 eV and 537.5 eV are attributed to the Sb $3d_{5/2}$ and Sb $3d_{3/2}$, respectively, characteristics of nonvalent antimony[37].

**Sensitivity simulation and LBL assembly of antimonene.** To investigate the key point of antimonene for improving SPR sensor performance, a numerical simulation was performed to evaluate the effect of the antimonene thickness on the sensitivity of the SPR sensor. The sensitivity can be defined as $S = \triangle\theta/\triangle n$, which is the ratio of the change in the resonance angle to the change in the refractive index of analyte[38]. The electric field distribution is shown in Supplementary Fig. 2, in which further enhancment rather than immediate drop in the electric field is observed while four layers antimonene is used. Figure 4a shows the variation in sensitivity with respect to the refractive index of the sensing medium and the number of antimonene layers in the proposed SPR biosensor. The simulation results suggest that antimonene materials can greatly improve the sensitivity of the SPR sensor. Figure 4b shows the variation in the sensitivity of the antimonene-based miRNA SPR sensor concerning the antimonene layer when the refractive index of the sensing medium is $1.37 + \triangle n$. The sensitivity first increases to the maximum (171° RIU$^{-1}$) when the number of antimonene layer is 4 and then begins to decrease when $L > 4$. The highest sensitivity was obtained with four layers of antimonene (details shown in Supplementary Information).

The above simulations provide clear guidance for the assembly of antimonene nanosheets on SPR chip surfaces. Experimentally, we assembled the antimonene nanosheets on a gold chip surface using a layer-by-layer technique. Figure 4c shows the representative AFM topography of antimonene nanosheets assembled on an Au chip surface. Although the surface of Au film exhibits rough, the distribution of the nanosheets is relatively uniform, and the average thickness of the nanosheets is ~5 nm (see Supplementary Fig. 2e). The consecutive build-up of the layer-by-layer (LBL) antimonene film was monitored by AFM (Supplementary Fig. 2). The increase in the surface coverage and thickness as a function of the number of assembled layers indicates that a very uniform increase in the average layer thickness occurred during each dipping cycle. To further confirm the results of the antimonene assembly, contact angle experiments were carried out (Fig. 4d). After the antimonene nanosheets were assembled on the sensor chip, the contact angle was approximately 38° due to the high hydrophilicity of the antimonene material. The long-time chemical durability of antimonene nanosheets is outstanding (shown in Supplementary Fig. 3). Hence, we can control the assembly of antimonene films on SPR chip surfaces using layer-by-layer assembly techniques.

**miRNA sensing performance of antimonene.** We measured the angle-resolved SPR spectra of the target miRNA-21 at very low concentration. The SPR responses upon the addition of complementary miRNA-21 are displayed in Fig. 5a. A prominent shift can be observed in the resonance angle, revealing the desorption of

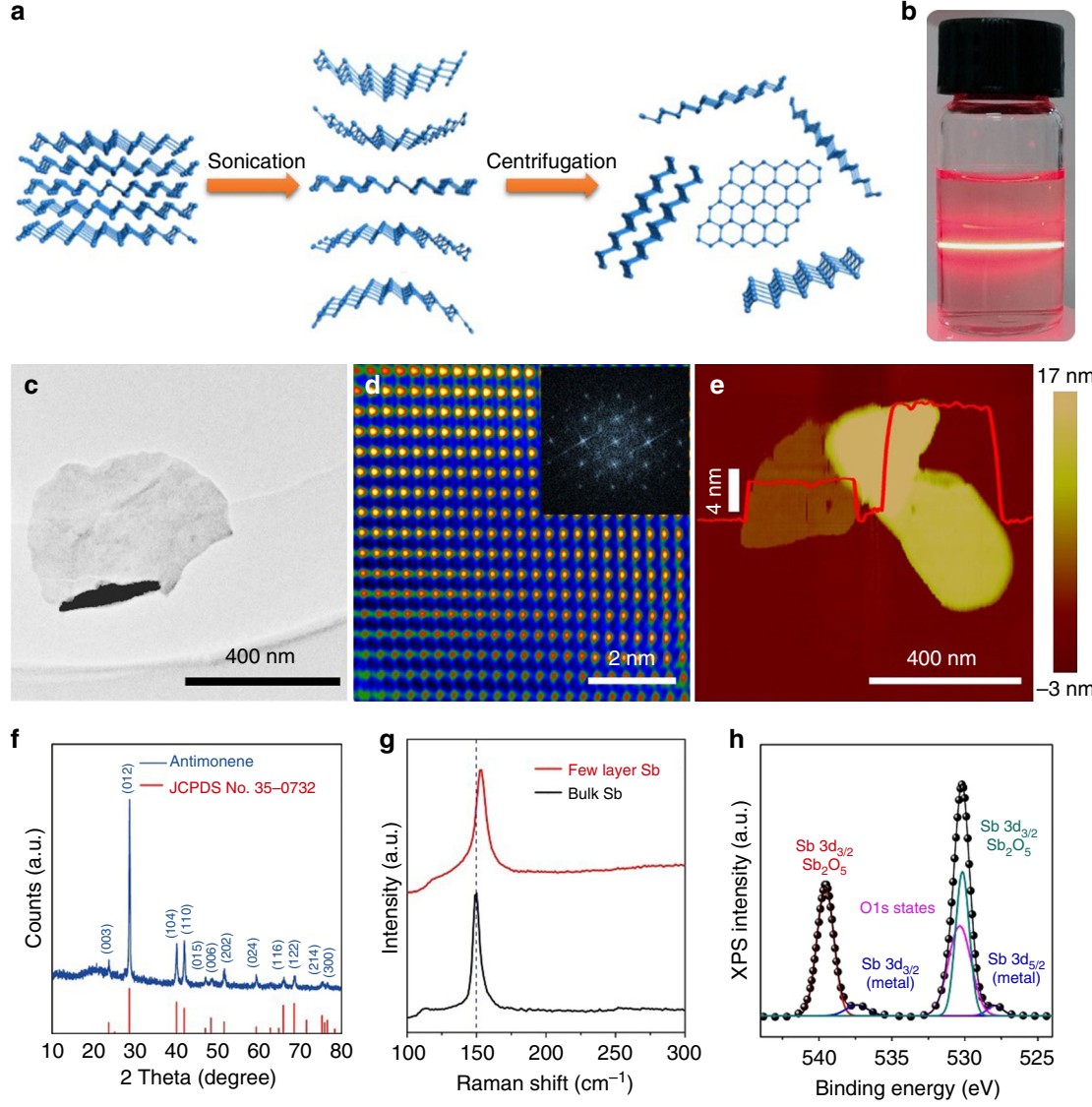

**Fig. 3** Fabrication and characterization of antimonene materials. **a** A schematic representation of the preparation process of two-dimensional antimonene. **b** Photograph of a dispersion of exfoliated antimonene showing the Faraday–Tyndall effect. **c**, **d** TEM (scale bar = 400 nm) and FFT-masked HRTEM images (scale bar = 2 nm) of few-layer antimonene after exfoliation. **e** AFM topography showing few-layer antimonene on mica (scale bar = 400 nm). **f** XRD spectrum of antimonene (blue line). **g** Raman spectra of bulk antimony with β-phase (black line) and few-layer antimonene (blue line). The two peaks represent two different vibrational modes. **h** XPS spectra of Sb 3d

AuNR-ssDNA. The hybridization of the target miRNA-21 results in an obvious left-shift in the SPR angles even at very low concentrations of $10^{-17}$ M. Using the same process but with unmodified-ssDNA, the shift in the SPR angle is barely detectable with the same concentrations (Fig. 5b). The calibration curve of the SPR angle shift versus the miRNA-21 concentration is shown in Fig. 5c. In the case of AuNR-ssDNA adsorption by antimonene, the detection limit of miRNA-21 was determined to be 10 aM according to the IUPAC guideline of a 3:1 signal to noise ratio. This detection limit is $10^5$ times lower than that using the non-modified ssDNA. Figure 5d shows the real-time desorption process of AuNR-ss DNA, which reaches a plateau after 5 min (Fig. 5d). When one base mismatched miRNA was used, the SPR angle slightly shifted to the right relative to that of AuNR-ssDNA (Fig. 5e), indicating good selectivity. Thus, opposite signals are obtained for mismatched miRNA, indicating that mismatched miRNA also bound the antimonene surface instead of binding to the AuNR-ssDNA conjugates[39,40]. Importantly, we observed similar results in the detection of miRNA-155 and ssDNA (Supplementary Fig. 4, 5),

suggesting that the antimonene nanomaterials are universal for the detection of miRNA and ssDNA.

To further demonstrate the sensitivity of our device, we compared the LOD of the antimonene 2D materials with that of previously reported miRNA biosensors[41–47] as presented in Fig. 5f. Our antimonene-based miRNA SPR biosensor outperforms other miRNA biosensors based on conventional 2D nanomaterials[41–47]. In particular, the LOD of miRNA sensing can approach 10 aM (~30 molecules for a 5 μL sample), indicating the great potential of antimonene nanosheets in applications, such as single-molecule biological imaging, clinical therapy, and environmental monitoring[48,49].

## Discussion
The unprecedented high sensitivity of the SPR sensor not only relies on the strong interaction between antimonene and single-stranded DNA but also benefits from the enhanced coupling between the localized-SPR (LSPR) of the gold nanorods and

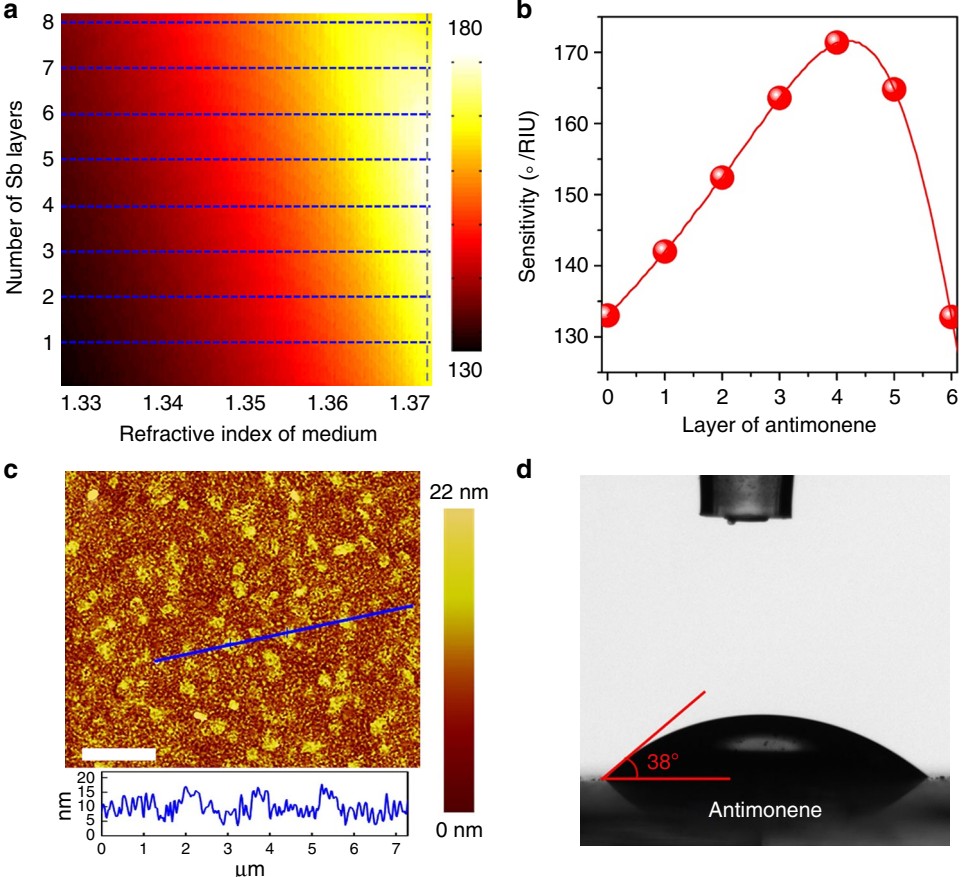

**Fig. 4** Simulation of antimonene SPR sensors and the antimonene assembly on the sensor chip. **a, b** The variation in the sensitivity of the proposed biochemical sensor when the refractive index of the sensing medium is $1.37 + \triangle n$ with respect to the different number of antimonene layers. To vividly illustrate the relationship of sensitivity with the number Sb layers, we assume that the number of Sb layers ($n = 2.1 + 0.45i$) can be continuously changed. **c** AFM images of antimonene on Au film (scale bar = 2 μm). **d** Images of distilled water droplets on antimonene assembled on Au film

propagating-SPR of the gold film. The preparation of AuNRs is characterized by TEM images (Fig. 6a). As shown in Fig. 6b, AuNRs modified with ssDNA show red-shifted plasmon bands at 513 and 747 nm. The signal enhancement due to the electromagnetic field coupling between the plasmonic properties of the AuNRs and propagating plasmons was expected. As shown in Fig. 6c–e, the electromagnetic field intensity distributions of single AuNRs on gold films with a 5 nm thick antimonene spacer were calculated by the finite-difference time-domain (FDTD) method[50]. The FDTD simulation parameters were consistent with the experimental conditions[51]. Compared with conventional SPR using gold film, a considerable electromagnetic enhancement was observed based on the evanescent field excitation and LSPR of the AuNRs with certain thickness of antimonene nanosheets. The simulation results of the local electric field distribution around antimonene nanosheets with different thicknesses were shown in Supplementary Fig. 6, which further verify the trend of sensitivity shown in Fig. 4b. The electric field at the optimal gap (antimonene layer) between the gold film and gold nanorod is increased by ~300 times at incident light wavelengths of 632.8 nm, where $|E| = |E_{local}/E_{in}|$, $E_{local}$ and $E_{in}$ are the local and incident electric fields, respectively. Thus, the sensor sensitivity is significantly improved by AuNRs.

In this paper, we demonstrated the efficiency and ultra-sensitivity of the antimonene-based SPR sensor in the quantitative detection of cancer-associated miRNA. Specifically, we applied the sensor to detect miRNA-21 and miRNA-155, which are promising biomarkers for cancer diagnosis. This antimonene-

based biosensor has a LOD of 10 aM, representing the highest sensitivity described thus far in miRNA detection based on direct detection and quantification of miRNA levels. More importantly, the signal amplification of the AuNRs and the interaction between antimonene and ssDNA/dsDNA were computationally and experimentally investigated. Consequently, the proposed biosensor represents the first methodology reported using antimonene materials for clinically relevant nucleic acid detection and constitutes an extraordinary opportunity for the development of lab-on-chip platforms. Nevertheless, for clinical and practical applications of the antimonene-based SPR sensor to be successfully used in early cancer diagnosis and the realization of point-of-care systems, future investigations of the specificity and high throughput are critically needed.

## Methods

**First-principles calculations**. Calculations were performed using DFT-based plane-wave pseudopotential methods as implemented in the Vienna Ab initio Simulation Package[52,53]. We described the electron-ion interactions using the projected augmented wave pseudopotentials with $5s^2 5p^3$ for Sb, $2s^2 2p^2$ for C, $2s^2 2p^3$ for N, $1s$ for H, and $2s^2 2p^4$ for O as valence electrons[54]. The generalized gradient approximation formulated by Perdew, Burke, and Ernzerhof was used as the exchange correlation functional[55]. To simulate DNAs absorbing antimonene, a separation of 20 Å in the $z$ direction was adopted to avoid interactions between adjacent antimonenes (in $6 \times 6$ or larger supercells). The nucleobase molecules were terminated with a methyl group to replace the sugar ring and generate an electronic environment as similar to a DNA chain as possible. The kinetic energy cut off for wave function expansion was set to 520 eV, and the single Γ point of the supercell was used for sampling the electronic Brillouin zone. Equilibrium structures of DNAs absorbing antimonene were obtained through total energy minimization with the energy convergence threshold of 1 meV.

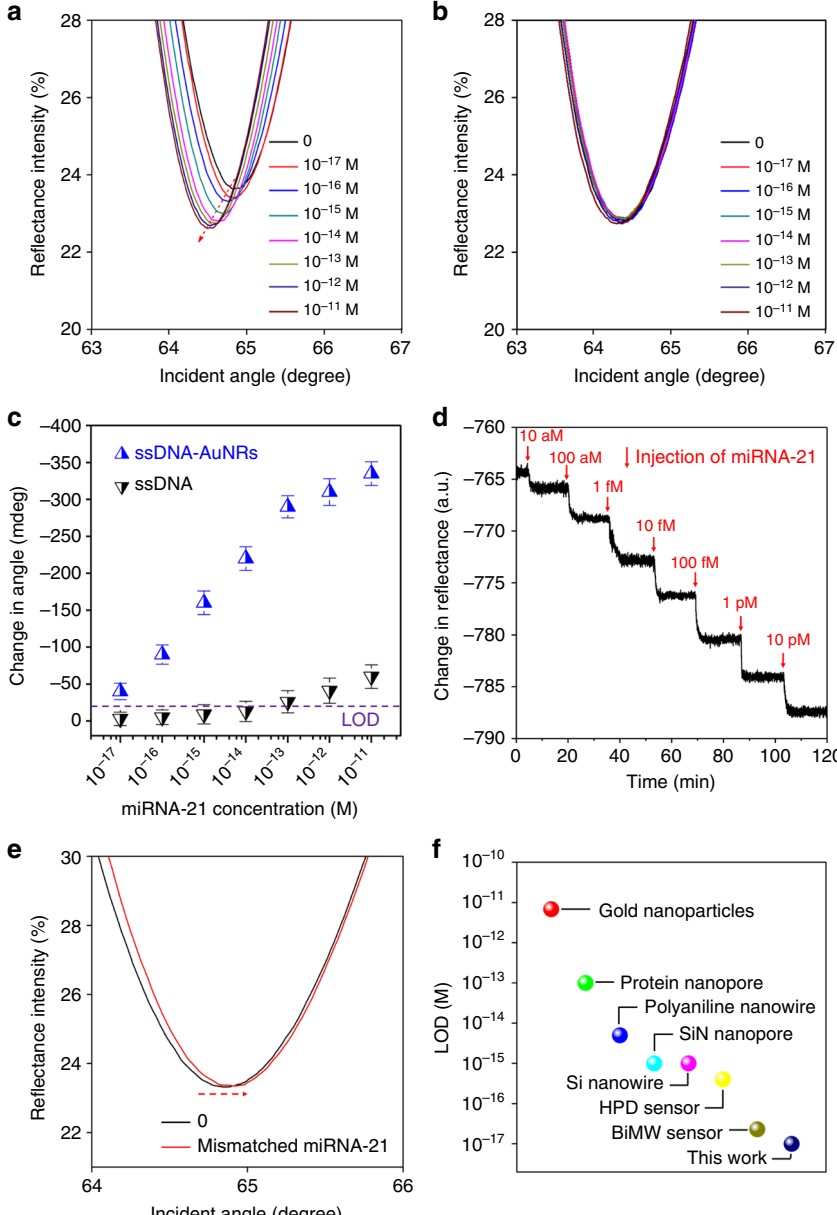

**Fig. 5** Sensing miRNA-21 using an antimonene SPR sensor. **a** SPR spectra with miRNA-21 concentrations ranging from $10^{-17}$ to $10^{-11}$ M using AuNRs amplification. The arrow denotes the shift in the SPR angle. **b** SPR spectra with miRNA-21 concentrations ranging from $10^{-17}$ to $10^{-11}$ M without AuNRs. **c** The relationship between the SPR angle and miRNA concentration. Each point corresponds to the SPR angle shift with the indicated miRNA concentration. All error bars is the standard error of SPR angle shift from five data points. **d** The real-time SPR response of ssDNA-AuNR desorption from the antimonene surface. **e** The SPR curve change of miRNA-21 contained one mismatched nucleobase (red line). **f** Comparison of the LOD of the antimonene miRNA SPR sensor with that of state-of-the-art sensors

We used the vdW-optB86b functional to properly take into account the long-range van der Waals interactions[56].

**FDTD simulation**. The near-field distribution of AuNRs on gold film under a 632.8 nm incident laser was simulated by the FDTD method. To model the AuNRs on a gold film, the FDTD method was used with antisymmetric and symmetric boundary conditions at the $x$–$y$ axis and $z$–$y$ axis, respectively. The propagation of the plane waves was directed along 45° from the $x$ axis. For all simulations, the parameters of the AuNRs nanostructures were set according to the average sizes (70 nm) measured from the experimental results. The mesh size was 1 nm. The electromagnetic field distribution of the Au nanorod was calculated at incident light wavelengths of 632.8 nm. The refractive index of Au used for the simulation was taken from the source program.

**Synthesis of antimonene nanosheets**. The antimonene nanosheets were prepared by probe-sonication liquid-phase exfoliation in ethanol. Pulverized antimony

(Sb) powder at an initial concentration of 30 mg mL$^{-1}$ was dispersed in a glass vial containing ethanol. Subsequently, the Sb powder solution was sonicated for 1 h in an ice-bath at 450 W and 22 kHz with ultrasound probe 0.5 s pulses. Then, the resulting solution was centrifuged at 1509.3×$g$ for 10 min. Finally, the supernatant containing the antimonene nanosheets were carefully collected in a clean glass vial for future use.

**Bioconjugation of AuNRs with ssDNA**. The AuNRs were chemically modified with 5′-thiol-called oligonucleotides according to the procedure described by Mirkin et al.[57]. In total, 25 μL of a 100 nM HS-ssDNA solution were added to 200 μL of the AuNRs solution (20 nM in 0.1 M PBS). After 16 h, the solution was mixed with 0.25 mL of 10% NaCl. Then, AuNR-ssDNA was centrifuged twice at 3018.6×$g$ for 20 s to remove the excess HS-ssDNA, and the particles were redispersed in PBS buffer (1 M NaCl, 100 mM PBS, pH = 7). The resultant colloidal solution was sonicated for 5 min and then stirred for 1 h at room temperature.

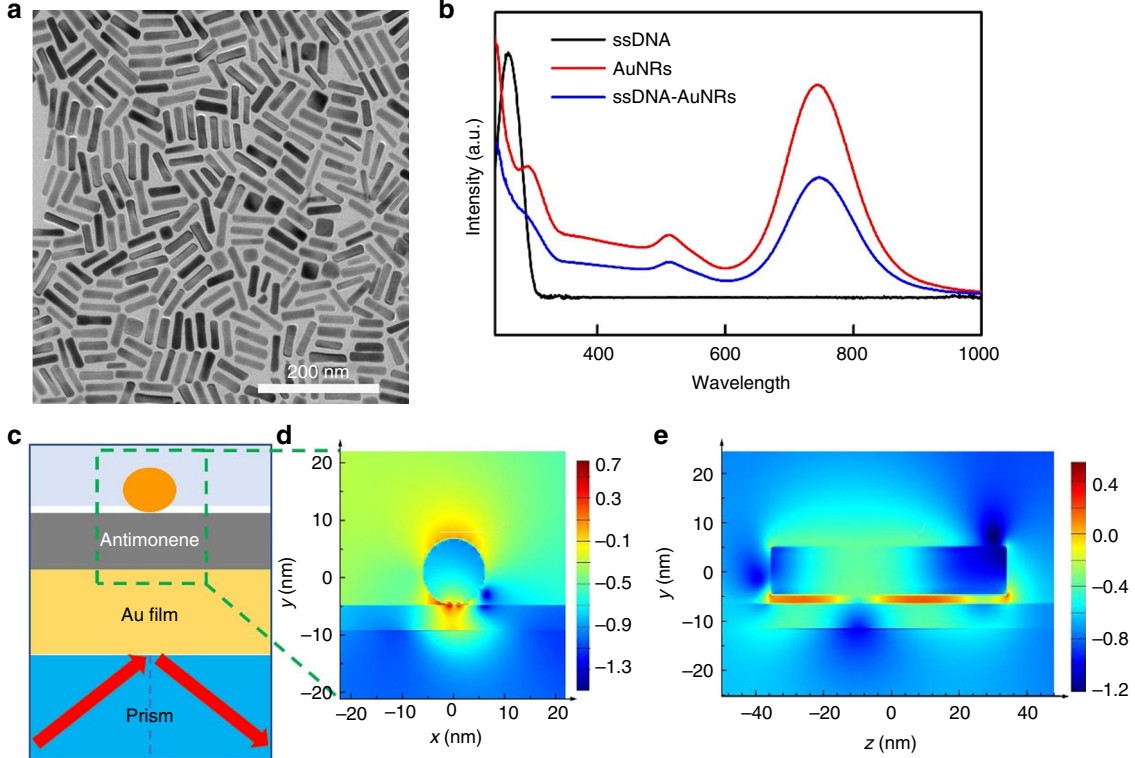

**Fig. 6** Signal amplification of AuNRs. **a** TEM image of gold nanorods (scale bar = 200 nm). **b** UV-Vis characterization spectra of ssDNA (black line), AuNRs (red line), and AuNR-ssDNA (blue line). **c** Schematic diagram of the SPR-AuNR configuration used for the FDTD simulation. **d** The FDTD calculated enhancement in the local electric field distribution ($\log |E E_{inc}^{-1}|^2$) of AuNRs at 632.8 nm with the incident wave-plane polarized along the x-direction. A 5 nm antimonene is set between the gold film and AuNRs. **e** The side view of FDTD calculated enhancement in the local electric field distribution

**Sensing test**. The AuNR-ssDNA solution was injected over the antimonene SPR chip and washed with PBS buffer. Complementary and noncomplementary miRNA in PBS were injected, and the hybridization signal was recorded. The sequences of the oligonucleotides are shown in Supplementary Table 1.

**Characterization**. SPR measurements were performed with a commercially available Time-Resolved Surface Plasmon Resonance Spectrometer (DyneChem, China). TEM and HRTEM images were obtained under a JEM-3200FS microscope (JEOL, Japan). The Raman spectra were collected using an iHR 320 spectrometer (Horibai, Japan). The AFM images were taken under an L01F4C8 microscope (Bruker, Germany). The UV-visible spectroscopy was performed using Cary60 (Agilent, Malaysia). XRD was determined by a D8 Advance instrument (Bruker, Germany). The XPS data were collected using an ESCALAB 250Xi XPS spectrometer (Thermo Fisher, America). The contact angle experiments were performed with a Theta instrument (Biolin Scientific, Sweden).

**Supporting information**. Details of the AFM images, SPR results for miRNA-155 and oligonucleotide sequences are provided. This material is available free of charge via the Internet at http://www.nature.com/naturecommunications

## Data availability
The authors declare that the data supporting the findings of this study are available within the paper and its Supplementary Information files.

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

## Acknowledgements

We acknowledge support from the National Natural Science Foundation of China (51602305, 51571100, 61722403, 51601131 and 61875139), the Science and Technology Innovation Commission of Shenzhen (JCYJ20170818141429525, JCYJ20170818141407343, JCYJ20170818141519879), the China Postdoctoral Science Foundation (2018M633118, 2017M620383, 2018M633102, 2018M633127), the Shenzhen Nanshan District Pilotage Team Program (LHTD20170006) and ARC (IH150100006, FT150100450, and CE170100039). Q. Bao acknowledges support from the Australian Research Council (ARC) Centre of Excellence in Future Low-Energy Electronics Technologies (FLEET). Calculations were performed in part at High Performance Computing Center of Jilin University.

## Author contributions

Q.B. and H.Z. conceived of the original concept. Q.B., H.Z., X.C. and L.Z. supervised the project. T.X. planned the project and performed most of the experiments. L.Z., Y.L. and Y.S. contributed to first-principles simulation. W.L., T.X. and K.Q. contributed to material preparations and characterizations. T.X., Y.X., and L.W contributed to the biosensing measurements. Z.D. and Y. Z. contributed to the FDTD measurements. T. X, B. S., Y. D., H. Z., X. C., L. Z. and Q. B. analyzed the data and co-wrote the paper. All authors discussed the results and commented on the manuscript. All authors have approved the final version of the manuscript.

## Additional information

**Competing interests:** The authors declare no competing interests.

