## [Peer Review File · Nature Communications]

Reviewers' Comments:

Reviewer #1:

Remarks to the Author:

In this manuscript, the authors present an SPR -based biosensor for miRNA detection using antimonene as an intermediate layer to capture and release Au-NR conjugated miRNA. The authors show that single stranded DNA has a higher affinity to the antimonene than the double stranded based on numerical calculations. To use this phenomenon for sensing, they first coat a standard SPR chip with antimonene flakes and Au-NR conjugated single-stranded DNA specific to the target miRNA. Then, the angular resonance shift measured on SPR set-up is quantified as numerous concentrations of target miRNA is inserted removing Au-NRs from the surface. The presented work is sizable. However this reviewer has concerns that the work applies a previously published technique (Graphene-based Portable SPR Sensor for the Detection of Mycobacterium tuberculosis DNA Strain. doi:10.1016/j.proeng.2016.11.520) by changing graphene with antimonene to enhance the sensitivity.

Below follows the argumentation as well as concerns that would need to be addressed:

- The title claims "single molecule" detection, however the presented results show no single molecule detection. It is claimed that the detection of 10 aM miRNA concentration corresponds to a single molecule level. It should be specified what sample volume was used for the measurements. For example, 10 aM concentration translates to ~6000 molecules for a 1 ml sample volume.
- It's claimed that increasing the number of layers of antimonene increases SPR sensitivity and the maximum sensitivity is reached with 4 layers. It is not clear why this is the case. Increasing the distance further from the metal surface should rather reduce the SPR sensitivity because the field strength decays.
- Antimonene nanosheets are claimed to be uniform, however Figure 4C shows significant variation in the surface profile of at least 22 nm, which corresponds to at least 6-7 layers of antimonene. The authors should comment on this point.
- The authors claim that miRNA with a mismatch in a single base pair does not bind with ssDNA but gets adsorbed on the sensor surface. Is it really the case? Wouldn't a miRNA strand with a single mismatch still bind to ssDNA?
- In the second paragraph of discussion, the authors claim "ultrasensitivity of the antimonene-based SPR sensor". However, it seems that the major sensitivity improvement of the SPR technique comes from the NR enhancement and not antimonene.
- In Figure 4A, it is not clear why the data along the y-axis related to the number of antimonene layers is continuous and not discrete. The authors should comment on this point.
- In Figures 5A and 5B authors should specify what kind of smoothening was used to present the SPR curves. Figure 5A has no noise, while SPR curves in 5B have noise.
- In Figures 5C and 5D the units on the y-axis are not clear. Are the units same or different? For instance in Figure 5D, the samples have 8 orders of magnitude difference in the concentration. However, the signals do not scale. The detection of 1 aM sample is presented but its signal is higher than the signal obtained from 10 aM sample. This looks confusing.
- Supplementary Figure S1 should have a scale bar.
- In the 2nd paragraph of the introduction it's claimed that the quantitative real-time PCR has "low sensitivity", which is not the case.

Reviewer #2:

Remarks to the Author:

Authors have investigated through numerical simulations and fabricated antimonene-based sensors for the detection of miRNA for early detection of cancer disease.

The paper is interesting, but I would like authors to address some comments as below.

At the beginning, authors investigate through DFT simulations the variation of the work function of antimonene (and graphene), when basis are deposited on top. I would like to see the dependence of the work-function variation with different orientations of the basis: are the obtained results the same?

Regarding the fabrication, I am wondering how much antimonene is stable. Is the reflectance for example changing as a function of time, while considering long time for the experiments?

Authors focus on sensitivity, but never mentioned selectivity. How good is the proposed sensor in sensing the target molecules, when other molecules are poured on top of it?

Given this I suggest to perform major revision to the article.

Reviewer #3:

Remarks to the Author:

I have read with interest the paper "Single-molecule detection of miRNA with an antimonene-based surface plasmon resonance sensor," Tianyu Xue, et al., which introduces a new kind of sensor specially design for sensing a specific cancer marker. In particular, the sensor is based on a 2D material, antimonene, which strongly interacts with a cancer marker (miRNA) present in a series of some of the most common cancers. The authors demonstrate better performance than the current state-of-the-art in terms of sensibility, namely more than a thousand times enhancement of the sensor sensitivity as compared to that of a similar sensor based on graphene. The paper addresses a timely topic of particular practical and societal importance, is well written, and the new findings are clearly identified, explained, and discussed. Therefore, I recommend the publication of the paper but I ask the authors to consider the following issues.

- The authors should comment on the generality of their proposed approach. In particular, can their sensor be used to detect other molecules, too?
- The authors should explain why is there an optimum number of layers (4) at which maximum sensitivity is achieved?
- It is not clear to this reader how the authors obtained the data presented in Figures 4a and 4b. Did they model a single layer of antimonene (or more) using FDTD? If the answer is yes, what value of the index of refraction did they used (the index of refraction of a single layer of a 2D material is not well defined).
- The field enhancement plotted in Figure 6 should not be presented on a log scale. Also, I suggest the authors plot $|E/E_{inc}|$ not $\log(|E|^2)$ as the former quantity has a more immediate physical meaning.
- The caption of Figure 2 should be extended so that the sensing approach illustrated in this figure can be better understood.

Manuscript ID: NCOMMS-18-16035A

Manuscript title: "Single-molecule detection of miRNA with an antimonene-based surface plasmon resonance sensor"

Point-by-point responses to Reviewers' Comments

We are very grateful for the comments from the reviewers. These comments are very valuable for improving the quality of this paper. Revisions and responses to address your comments are presented as below.

Reviewer #1

In this manuscript, the authors present an SPR -based biosensor for miRNA detection using antimonene as an intermediate layer to capture and release Au-NR conjugated miRNA. The authors show that single stranded DNA has a higher affinity to the antimonene than the double stranded based on numerical calculations. To use this phenomenon for sensing, they first coat a standard SPR chip with antimonene flakes and Au-NR conjugated single-stranded DNA specific to the target miRNA. Then, the angular resonance shift measured on SPR set-up is quantified as numerous concentrations of target miRNA is inserted removing Au-NRs from the surface. The presented work is sizable. However this reviewer has concerns that the work applies a previously published technique (Graphene-based Portable SPR Sensor for the Detection of Mycobacterium tuberculosis DNA Strain. doi:10.1016/j.proeng.2016.11.520) by changing graphene with antimonene to enhance the sensitivity.

REPLY: We thank the reviewer for identifying the key content of this work. However,

we disagree with the comment that “this reviewer has concerns that the work applies a previously published technique (Graphene-based Portable SPR Sensor for the Detection of Mycobacterium tuberculosis DNA Strain. doi:10.1016/j.proeng.2016.11.520) by changing graphene with antimonene to enhance the sensitivity”. In order to address the reviewer’s concern, we have cited this reference (Ref. 19) and amended the main text. We want to emphasize that our current work is essentially different from the previous work on the following aspects:

1. Stronger interaction. First-principles energetic calculations reveal that antimonene has substantially stronger interaction with single-stranded DNA than graphene that was previously used in DNA molecule sensing.

2. New material. Few-layer antimonene nanosheets prepared by liquid exfoliation were used as sensing media in this work. It is found that this new material delivers much better sensitivity than graphene, according to both simulation and experiments. The above literature is a good reference for comparison.

3. Sensing performance. We developed a novel SPR sensor by using antimonene materials enhanced by strong interaction with ssDNA and traced attomolar-level quantification of miRNA molecule. This antimonene-based biosensor, with a LOD of 10 aM, represents the highest sensitivity for miRNA detection so far described in direct detection and quantification of miRNA levels.

Given the above, we have demonstrated an efficient and ultrasensitive surface plasmon resonance sensor based on antimonene for the quantitative detection of cancer-associated miRNA. We believe this work have far-ranging impact and will

stimulate interest in several communities, e.g., layered 2D materials, biosensor devices, DNA and miRNA. Therefore, it will be ideally suited for the diverse readership of Nature Communications.

ACTIONS:

We cited the reference *Procedia Eng.* **168**, 541-545 (2016) as Ref. 19 in 1st paragraph of page 4 of revised manuscript.

To highlight novelty of our work, we added one sentence “This proposed methodology based on antimonene materials for nucleic acid detection holds intriguing potential for the development of multiplexed lab-on-chip platforms, which can be further applied for clinical purposes.” in INTRODUCTION part of 5th page in the revised manuscript.

And we also change “First-principles energetic calculations reveal that antimonene has substantially stronger interaction with single-stranded DNA than the graphene that has been previously used in DNA molecule sensing” into “First-principles energetic calculations reveal that antimonene has substantially stronger interaction with single-stranded DNA than the graphene that has been previously used in DNA molecule sensing thanks to more delocalized 5s/5p orbitals of antimonene.” in 1st paragraph of page 2 of revised manuscript.

Comment 1: The title claims “single molecule” detection, however the presented results show no single molecule detection. It is claimed that the detection of 10 aM miRNA concentration corresponds to a single molecule level. It should be specified

what sample volume was used for the measurements. For example, 10 aM concentration translates to ~6000 molecules for a 1 ml sample volume.

REPLY: Thank for raising this good point. For the miRNA solution with a concentration of 10 aM, there are around ~30 molecules in a 5 μ L sample. We have addressed this in the revised manuscript. The new reference has been cited in the revised manuscript.

48. Zhang, X., Liu, C., Sun, L., Duan, X. & Li, Z., Lab on a single microbead: an ultrasensitive detection strategy enabling microRNA analysis at the single-molecule level. *Chem. Sci.* **6**, 6213-6218 (2015).
49. Neely, L.A. et al. A single-molecule method for the quantitation of microRNA gene expression. *Nat. Meth.* **3**, 41 (2005).

ACTIONS:

In the 2nd paragraph of page 11 in the revised manuscript, we changed the sentence “In particular, the LOD of miRNA sensing can approach 10 aM, reaching the single-molecule level” into. “In particular, the LOD of miRNA sensing can approach 10 aM (~30 molecules for a 5 μ L sample), reaching the single-molecule level.^{48,49}”.

We also cited the reference new references 48, 49 in 2nd paragraph of page 11 of revised manuscript.

Comment 2: It's claimed that increasing the number of layers of antimonene increases SPR sensitivity and the maximum sensitivity is reached with 4 layers. It is

not clear why this is the case. Increasing the distance further from the metal surface should rather reduce the SPR sensitivity because the field strength decays.

REPLY: Thank you very much for the good comments. In traditional SPR model (gold film or silver film), the SPR sensitivity for analyte will be reduced with increasing the distance from the metal surface due to reduced local electrical field. However, in this novel SPR sensor, due to the appearance of antimonene which has a large refractive index ($2.1+0.45i$) between gold film and gold nanorod, the maximum local electrical field happens at the surface of antimonene nanosheet with certain thickness. To verify the optimal thickness in figure 4b, we performed more FDTD simulation to reveal the local electric field distribution as a function of the thickness of antimonene nanosheet, as shown in the revised figure S6. We can observe that the electric field is the strongest when the antimonene thickness is 5 nm, which consequently deliver the best sensitivity.

In addition, we have also simulated the profile of electric field crossing gold thin film and 5 nm antimonene nanosheet using MATLAB software. Unlike other sensing materials which cause the immediate decrease of electrical field, the incorporation of antimonene nanosheet onto gold thin film will further increase the electrical field until a maximum at the outer surface of antimonene nanosheet. These results suggest the key role of antimonene for electric field enhancement at the interface between gold film and sensing medium.

ACTIONS:

In page 8 of the revised Supplementary information, we have added new figure S6.

Figure S6. The local electric field distribution around AuNRs placed on antimonene nanosheets with different thicknesses. a, A antimonene film with the thickness of 1 nm is placed between the gold film and AuNRs. **b,** A antimonene film with the thickness of 5 nm is placed between the gold film and AuNRs. **c,** A antimonene film with the thickness of 7 nm is placed between the gold film and AuNRs. **d,** A antimonene film with the thickness of 13 nm is placed between the gold film and AuNRs. Based on above simulation, it can be seen that the local electric field is the strongest when the antimonene thickness is 5 nm.

Figure. The electric field distribution of the proposed SPR sensor based on antimonene with 5 nm thickness.

In the 1st paragraph of page 12 in the revised manuscript, we added sentence “The simulation results of the local electric field distribution around antimonene nanosheets with different thicknesses were shown in Figure S6, which further verify the trend of sensitivity shown in Figure 4b.”

Comment 3: Antimonene nanosheets are claimed to be uniform, however Figure 4C shows significant variation in the surface profile of at least 22 nm, which corresponds to at least 6-7 layers of antimonene. The authors should comment on this point.

REPLY: Thanks for the comments. The relatively large scale bar of 22 nm in the AFM is caused by the roughness of gold film, which is the substrate for antimonene nanosheets. We have normalized the scale bar, and replotted the AFM image by added

the sectional profile. The sectional view in Figure 4c shows the thickness of antimonene nanosheets is around 5 nm, in good agreement with the thickness of antimonene in Figure 3e.

ACTIONS:

In the 1st paragraph of page 10 in the revised manuscript, we have changed the sentence “The distribution of the nanosheets is relatively uniform, and the average thickness of the nanosheets is approximately 5 nm (see Supplementary Information Figure S1e).” into “Although the surface of Au film exhibits rough, the distribution of the nanosheets is relatively uniform, and the average thickness of the nanosheets is approximately 5 nm (see Supplementary Information Figure S2e).”

And we have changed Figure 4C in the revised manuscript.

Figure 4c. AFM topography of antimonene nanosheets on Au film.

Comment 4: The authors claim that miRNA with a mismatch in a single base pair

does not bind with ssDNA but gets adsorbed on the sensor surface. Is it really the case?

Wouldn't a miRNA strand with a single mismatch still bind to ssDNA?

REPLY: Thanks for the comments. The miRNA with a mismatched in a single base will not bind with probe ssDNA. This phenomenon has been systematically investigated in previous works by other researchers. To address this point more clearly, we added the following two references in the reference list:

39. Dong, X., Shi, Y., Huang, W., Chen, P. & Li, L.J., Electrical Detection of DNA Hybridization with Single-Base Specificity Using Transistors Based on CVD-Grown Graphene Sheets. *Adv. Mater.* **22**, 1649-1653 (2010).
40. He, S. et al. A Graphene Nanoprobe for Rapid, Sensitive, and Multicolor Fluorescent DNA Analysis. *Adv. Funct. Mater.* **20**, 453-459 (2010).

ACTIONS:

In the 1st paragraph of page 11 in the revised manuscript, we have changed the sentence "...When mismatched miRNA was used, the SPR angle slightly shifted to the right relative to that of AuNR-ssDNA (Figure 5e)." into "...When one base mismatched miRNA was used, the SPR angle slightly shifted to the right relative to that of AuNR-ssDNA (Figure 5e)."

We also cited the references as 39, 40 in the 1st paragraph of page 11 in the revised manuscript.

Comment 5: In the second paragraph of discussion, the authors claim

“ultrasensitivity of the antimonene-based SPR sensor”. However, it seems that the major sensitivity improvement of the SPR technique comes from the NR enhancement and not antimonene.

REPLY: Thanks for the comments. The nanorods enhancement is important to the sensitivity improvement to single-molecule level. However, the antimonene nanosheets play a major role to achieve the ultrahigh sensitivity. Firstly, we find that antimonene has much better sensitivity than graphene that was previously used in DNA molecule sensing by DFT. The underlying mechanism is related to the more delocalized 5s/5p orbitals or the buckling honeycomb lattice of antimonene. Secondly, the physical property of antimonene materials, refractive index is $2.1+0.45i$, can greatly improve the sensitivity of the SPR sensor by simulation. Lastly, it is also interesting to find that the chemical interactions of antimonene with single-stranded DNA and double-stranded DNA are significant different. Therefore, the antimonene is essential and vital for ultrasensitive detection of miRNA.

Comment 6: In Figure 4A, it is not clear why the data along the y-axis related to the number of antimonene layers is continuous and not discrete. The authors should comment on this point.

REPLY: Thank the reviewer for pointing out inappropriate presentation. Theoretically, we assume that the number of antimonene layers versus refractive index can be continuously changed for the ease of simulation and data presentation. Practically, we only refer to those results which exact correspond to the integral number of layers. In

order to avoid mis-understanding, we have added dashed lines in Figure 4a to indicate the refractive index as well as sensitivity that are meaningful.

ACTIONS:

In the 1st paragraph of page 25 in the revised manuscript, we added one sentence: “Theoretically, we assume that the number of Sb layers versus refractive index can be continuously changed for the ease of simulation. Practically, we only refer to those results which exact correspond to the integral number of layers.”.

And we also change the Figure 4a in the revised manuscript.

Figure 4a, The variation in the sensitivity of the proposed biochemical sensor when the refractive index of the sensing medium is $1.37 + \Delta n$ with respect to the different number of antimonene layers. Theoretically, we assume that the number of Sb layers versus refractive index can be continuously changed for the ease of simulation. Practically, we only refer to those results which exactly correspond to the integral

number of layers.

Comment 7: In Figures 5A and 5B authors should specify what kind of smoothing was used to present the SPR curves. Figure 5A has no noise, while SPR curves in 5B have noise.

REPLY: We thank the reviewer for identifying this point. c We agree with the reviewer that the SPR curves in this paper should be smoothed by Savitzky-Golay method. We have smoothed the curves in figure 5B using the same smoothing function (*i.e.*, $x_{k,\text{smooth}} = \bar{x}_k = \frac{1}{H} \sum_{i=-w}^{+w} x_{k+i} + h_i$) as that performed for Figure 5A, which was supplied by Savitzky-Golay method (*i.e.*, $x_{k,\text{smooth}} = \bar{x}_k = \frac{1}{H} \sum_{i=-w}^{+w} x_{k+i} + h_i$), as shown in figure 5b.

ACTIONS:

In the figure 5 of the page 26 in the revised manuscript, we have refreshed Figure 5b.

Figure 5. Sensing miRNA-21 using an antimonene SPR sensor. a, SPR spectra with miRNA-21 concentrations ranging from 10⁻¹⁷ to 10⁻¹¹ M using AuNRs

amplification. The arrow denotes the shift in the SPR angle. **b**, SPR spectra with miRNA-21 concentrations ranging from 10^{-17} to 10^{-11} M without AuNRs.

Comment 8: In Figures 5C and 5D the units on the y-axis are not clear. Are the units same or different? For instance, in Figure 5D, the samples have 8 orders of magnitude difference in the concentration. However, the signals do not scale. The detection of 1 aM sample is presented but its signal is higher than the signal obtained from 10 aM sample. This looks confusing.

REPLY: Thanks for good comments which we completely agree. We recorded the SPR spectrum curves after 15 min of injection of different concentrations (from 10 aM to 10 pM), respectively. For each detected concentration, the miRNA molecules exhibit saturated adsorption process. We can obtain a good linear relation from the calibration curve as the x-axis presented on a log scale. To make it clear, we change the y-axis in figure 5c and the y-axis in figure 5d. The y-axis in Figure 5c and Figure 5d are different. The y-axis in Figure 5c represents the change of SPR spectra resonance angle. The y-axis in Figure 5d represents the change of relative reflectance intensity. The mistakes of target miRNA concentrations have been corrected. The updated Figure 5c and Figure 5d have been shown below.

ACTIONS:

In the figure 5 of the page 26 in the revised manuscript, we have updated Figure 5c and Figure 5d.

Comment 9: Supplementary Figure S1 should have a scale bar.

REPLY: We appreciate the referee's comment. The AFM image has been updated. We have added the scale bars, vertical scale bars and root-mean-square roughness (RMS) in the revised Figure S2 in Supplementary information.

ACTIONS:

In the figure S2 of the page 4 in the revised Supplementary information, we have added the scale bar, color bar and root-mean-square roughness (RMS) in each figure.

Figure S2. AFM images and height profile of antimonene film deposited using the electrostatic layer-by-layer method. a-f, Antimonene thin films with varied numbers of layers from 0 layer to 5 layers. The scale bars are 3 μm .

Comment 10: In the 2nd paragraph of the introduction it's claimed that the quantitative real-time PCR has "low sensitivity", which is not the case.

REPLY: Thanks for the referee's comment. We have changed the description of RT-PCR in the revised manuscript.

ACTIONS:

In the 2nd paragraph of page 3 in the revised manuscript, we have changed the sentence "Traditionally, the use of miRNA detection techniques, such as quantitative

real-time PCR (qRT-PCR),⁶ northern blotting,⁷ and microarray-based hybridization,⁸ is limited in early diagnosis in clinical practice due to the high cost, complex operations and low sensitivity.” into “Traditionally, the use of miRNA detection techniques, such as quantitative real-time PCR (qRT-PCR),⁶ northern blotting,⁷ and microarray-based hybridization,⁸ is limited in early diagnosis in clinical practice due to the difficult amplification, the high cost, complex operations or low sensitivity.”

Reviewer #2

Authors have investigated through numerical simulations and fabricated antimonene-based sensors for the detection of miRNA for early detection of cancer disease.

The paper is interesting, but I would like authors to address some comments as below.**Comment 1:** At the beginning, authors investigate through DFT simulations the variation of the work function of antimonene (and graphene), when basis are deposited on top. I would like to see the dependence of the work-function variation with different orientations of the basis: are the obtained results the same?

REPLY: We thank the reviewer for raising this point. Accordingly, we have performed additional calculations of the nucleobases on top of antimonene/graphene with varied adsorption orientations. In particular, we calculated the single-stranded DNA (A nucleobases) on graphene and antimonene respectively, as well as the double-stranded DNA (A-T base-pairs) on antimonene with the different orientation angles changing from 0° , 30° , 60° , 90° , 120° , 150° , to 180° . The calculated adsorption energies of the three cases are shown in the newly added figure S1. We can see that the adsorption energies show tiny change with varied orientations of bases/base-pairs. The fact that antimonene exhibits the stronger interaction with DNAs is robust. According to the work function results, it is suggested that the maximum 2% change with different orientations for all the cases. These results indicate that the adsorption orientation of nucleobases have negligible effect on the conclusion reached by the DFT calculations.

Figure S1. The DFT calculation for the nucleobases on top of antimonene/graphene with varied adsorption orientations. We calculated the single-stranded DNA (A nucleobases) on graphene and antimonene respectively, and the double-stranded DNA (A-T base-pairs) on antimonene with the different orientation angles changing from 0°, 30°, 60°, 90°, 120°, 150°, to 180°. The calculated adsorption energies of the three cases are shown. According to the work function results, it is suggested that the maximum 2% change with different orientations for all the cases.

ACTIONS:

In the page 2 in the revised Supplementary Information, the calculated adsorption energies for the nucleobases on top of antimonene/graphene with varied adsorption

orientations are shown in the newly added figure S1.

In the 1st paragraph of page 6 in the revised manuscript, we added the pertinent discussion “Further calculations of adsorption energies and work functions for the nucleobases on top of antimonene/graphene with varied adsorption orientations indicate that the bases adsorption orientations have negligible effect on the above results obtained (see Figure S1).”

Comment 2: Regarding the fabrication, I am wondering how much antimonene is stable. Is the reflectance for example changing as a function of time, while considering long time for the experiments?

REPLY: Thanks for raising a very good point. The long time chemical durability of antimonene nanosheets was updated by surface plasmon resonance technique at the water solution for 2 h, 6h, 12h, and 24h, which showed negligible shift of SPR spectra (Figure 4). This indicates that this sensor chip materials are very stably for the sensing experiments. The new results as shown in below:

Figure S3. The SPR spectra for long time chemical durability of antimonene nanosheets. The SPR curves of antimonene-probe AuNR-ssDNA at the water solution for 2 h, 6h, 12h, and 24h.

ACTIONS:

We have added Figure S3 in page 4 in the revised Supplementary Information.

We also added sentence “The long-time chemical durability of antimonene nanosheets is outstanding (shown in Figure S3).” in 1st of page 10 in the revised manuscript.

Comment 3: Authors focus on sensitivity, but never mentioned selectivity. How good is the proposed sensor in sensing the target molecules, when other molecules are poured on top of it?

REPLY: Thanks for the good question. The selectivity of this novel sensor is

outstanding. When one base mismatched miRNA was used, the SPR angle slightly shifted to the right relative to that of AuNR-ssDNA. We have discussed the selectivity of this novel sensor in page 11, and the data was shown in Figure 5e and Figure S4b. The experiment phenomenon will be more obvious (SPR curves right shift), when other molecules are poured on top of it. To verify this, we obtained the data of mismatched miRNA-21 and miRNA-155.

ACTIONS:

In the 1st paragraph of page 11 in the revised manuscript, we have changed the sentence “When mismatched miRNA was used, the SPR angle slightly shifted to the right relative to that of AuNR-ssDNA (Figure 5e).” into “When one base mismatched miRNA was used, the SPR angle slightly shifted to the right relative to that of AuNR-ssDNA (Figure 5e), indicating good selectivity.”

Reviewer #3

I have read with interest the paper “Single-molecule detection of miRNA with an antimonene-based surface plasmon resonance sensor,” Tianyu Xue, et al., which introduces a new kind of sensor specially design for sensing a specific cancer marker. In particular, the sensor is based on a 2D material, antimonene, which strongly interacts with a cancer marker (miRNA) present in a series of some of the most common cancers. The authors demonstrate better performance than the current state-of-the-art in terms of sensibility, namely more than a thousand times enhancement of the sensor sensitivity as compared to that of a similar sensor based on graphene. The paper addresses a timely topic of particular practical and societal importance, is well written, and the new findings are clearly identified, explained, and discussed. Therefore, I recommend the publication of the paper but I ask the authors to consider the following issues.

Comment 1: The authors should comment on the generality of their proposed approach. In particular, can their sensor be used to detect other molecules, too?

REPLY: Thank for good point. This proposed approach can be used not only for miRNA-21 and miRNA-155 detection but also for ssDNA (biomarker). To further confirm the universality of our proposed approach, we tried ssDNA, of which base sequence is 5'-GCT AGA GAT TTT CCA CAC TGA CT-3' (see Supplementary Figure S5). The same phenomenon has been observed for this ssDNA target. To summarize, we have obtained repeatable and constant sensing performance from three kind of sample (miRNA-21, miRNA-155, ssDNA) using this antimonene-based

sensor.

Figure S5. Sensing of ssDNA using an antimonene SPR sensor. a, SPR spectra with ssDNA concentrations ranging from 10^{-17} to 10^{-12} M obtained using AuNR amplification. The arrow denotes the shift in the SPR angle. **b**, The relationship between the SPR angle and ssDNA concentration. Each point corresponds to an SPR angle shift for the indicated concentrations of ssDNA.

ACTIONS:

In the page 6 in the revised Supplementary Information, we have added the new figure S5.

In the 1st paragraph of page 11 in the revised manuscript, we have changed the sentence “Importantly, we observed similar results in the detection of miRNA-155 (Figure S3), suggesting that the antimonene nanomaterials are universal for miRNA and ssDNA detection.” into “Importantly, we observed similar results in the detection of miRNA-155 and ssDNA (Figure S4, S5), suggesting that the antimonene nanomaterials are universal for the detection of miRNA and ssDNA.”

Comment 2: The authors should explain why is there an optimum number of layers (4) at which maximum sensitivity is achieved?

REPLY: Thanks for the good comments, which are quite similar to the question No 3 raised by the first reviewer. We would like to address the reviewer's comments as below:

However, in this novel SPR sensor, due to the appearance of antimonene which has a large refractive index ($2.1+0.45i$) between gold film and gold nanorod, the maximum local electrical field happens at the surface of antimonene nanosheet with certain thickness. To verify the optimal thickness in figure 4b, we performed more FDTD simulation to reveal the local electric field distribution as a function of the thickness of antimonene nanosheet, as shown in the revised figure S6. We can observe that the electric field is the strongest when the antimonene thickness is 5 nm, which consequently deliver the best sensitivity.

In addition, we have also simulated the profile of electric field crossing gold thin film and 5 nm antimonene nanosheet using MATLAB software. Unlike other sensing materials which cause the immediate decrease of electrical field, the incorporation of antimonene nanosheet onto gold thin film will further increase the electrical field until a maximum at the outer surface of antimonene nanosheet. These results suggest the key role of antimonene for electric field enhancement at the interface between gold film and sensing medium.

ACTIONS:

In page 7 of the revised Supplementary information, we have added new figure S6.

Figure S6. The local electric field distribution around AuNRs placed on antimonene nanosheets with different thicknesses. a, A antimonene film with the thickness of 1 nm is placed between the gold film and AuNRs. **b,** A antimonene film with the thickness of 5 nm is placed between the gold film and AuNRs. **c,** A antimonene film with the thickness of 7 nm is placed between the gold film and AuNRs. **d,** A antimonene film with the thickness of 13 nm is placed between the gold film and AuNRs. Based on above simulation, it can be seen that the local electric field is the strongest when the antimonene thickness is 5 nm.

Figure. The electric field distribution of the proposed SPR sensor based on antimonene with 5 nm thickness.

In the 1st paragraph of page 13 in the revised manuscript, we added sentence “The simulation results of the local electric field distribution around antimonene nanosheets with different thicknesses were shown in Figure S6, which further verify the trend of sensitivity shown in Figure 4b.”

Comment 3: It is not clear to this reader how the authors obtained the data presented in Figures 4a and 4b. Did they model a single layer of antimonene (or more) using FDTD? If the answer is yes, what value of the index of refraction did they used (the index of refraction of a single layer of a 2D material is not well defined).

REPLY: Thanks for the comment. For the data presented in Figures 4a and 4b, we employ the transfer matrix method using MATLAB software to model the reflectance

of the incident TM-polarized light. In the proposed biosensor, all layers are stacked along the direction perpendicular to the prism, and each layer is defined by the thickness (d_k), refractive index (n_k), and dielectric constant (ϵ_k), respectively. The index of refraction was used as $2.1+0.45i$ for a single layer antimonene. The thickness of antimonene is 5 nm, which is agree with experiments. We have changed the caption of Figure 4. The change in the resonance angle ($\Delta\theta$) is caused by the change in the refractive index of the sensing medium (Δn), and the sensitivity can be defined as $S_R = \Delta\theta/\Delta n$.^[1-3]

Theoretically, we assume that the number of antimonene layers versus refractive index can be continuously changed for the ease of simulation and data presentation. Practically, we only refer to those results which exact correspond to the integral number of layers. To avoid mis-understanding, we have added dashed lines in Figure 4a to indicate the refractive index as well as sensitivity that are meaningful.

[1] W. H. Hansen, Electric fields produced by the propagation of plane coherent electromagnetic radiation in a stratified medium. *J. Opt. Soc. Am.* **58**, 380–388 (1968).

[2] R. Verma, B.D. Gupta, R. Jha, Sensitivity enhancement of a surface plasmon resonance based biomolecules sensor using graphene and silicon layers. *Sens. Actuators B* **160**, 623–631 (2011).

[3] P.K. Maharana, R. Jha, Chalcogenide prism and graphene multilayer based surface plasmon resonance affinity biosensor for high performance. *Sens. Actuators B* **169**, 161–166 (2012).

ACTIONS:

In the 1st paragraph of page 25 in the revised manuscript, we added one sentence: “Theoretically, we assume that the number of Sb layers versus refractive index can be continuously changed for the ease of simulation. Practically, we only refer to those results which exact correspond to the integral number of layers.”.

And we also change the Figure 4a in the revised manuscript.

Figure 4a, The variation in the sensitivity of the proposed biochemical sensor when the refractive index of the sensing medium is $1.37 + \Delta n$ with respect to the different number of antimonene layers. To vividly illustrate the relationship of sensitivity with the number Sb layers, we assume that the number of Sb layers ($n=2.1+0.45i$) can be continuously changed.

In the figure 4 of the page 25 in the revised manuscript, we changed the caption “**Figure 4. Simulation of antimonene SPR sensors and the results of the**

antimonene assembly on the sensor chip. a, b, The variation in the sensitivity of the proposed biochemical sensor when the refractive index of the sensing medium is $1.37 + \Delta n$ with respect to the different number of antimonene layers. **c,** AFM images of antimonene on Au film. **d,** Images of distilled water droplets on antimonene assembled on Au film.” into “**Figure 4. Simulation of antimonene SPR sensors and the results of the antimonene assembly on the sensor chip. a, b,** The variation in the sensitivity of the proposed biochemical sensor when the refractive index of the sensing medium is $1.37 + \Delta n$ with respect to the different number of antimonene layers. To vividly illustrate the relationship of sensitivity with the number Sb layers, we assume that the number of Sb layers ($n=2.1+0.45i$) can be continuously changed. **c,** AFM images of antimonene on Au film. **d,** Images of distilled water droplets on antimonene assembled on Au film.”.

Comment 4: The field enhancement plotted in Figure 6 should not be presented on a log scale. Also, I suggest the authors plot $|E/E_{inc}|$ not $\log(|E|^2)$ as the former quantity has a more immediate physical meaning.

REPLY: We appreciate the referee's comment. The Figure 6 has been updated as below.

Figure 6. Signal amplification of AuNRs. **a**, TEM image of gold nanorods. **b**, UV-Vis characterization spectra of ssDNA, AuNRs, and AuNR-ssDNA. **c**, Schematic diagram of the SPR-AuNR configuration used for the FDTD simulation. **d**, The FDTD calculated enhancement in the local electric field distribution ($\log |E/E_{inc}|^2$) of AuNRs at 632.8 nm with the incident wave-plane polarized along the x-direction. A 5 nm antimonene is set between the gold film and AuNRs. **e**, The side view of FDTD calculated enhancement in the local electric field distribution.

ACTIONS:

We have uploaded the Figure 6 in page 27 in the revised manuscript.

In the figure 6 of the page 27 in the revised manuscript, we changed the caption of Figure 6 “**Figure 6. Signal amplification of AuNRs.** **a**, TEM image of gold nanorods. **b**, UV-Vis characterization spectra of ssDNA, AuNRs, and AuNR-ssDNA. **c**, Schematic diagram of the SPR-AuNR configuration used for the FDTD simulation. The FDTD calculated enhancement in the local electric field distribution ($\log |E|^2$) of

AuNRs at 632.8 nm with the incident wave-plane polarized along the x-direction. A 5 nm antimonene is set between the gold film and AuNRs.” into “**Figure 6. Signal amplification of AuNRs.** **a**, TEM image of gold nanorods. **b**, UV-Vis characterization spectra of ssDNA, AuNRs, and AuNR-ssDNA. **c**, Schematic diagram of the SPR-AuNR configuration used for the FDTD simulation. **d**, The FDTD calculated enhancement in the local electric field distribution ($\log |E/E_{inc}|^2$) of AuNRs at 632.8 nm with the incident wave-plane polarized along the x-direction. A 5 nm antimonene is set between the gold film and AuNRs. **e**, The side view of FDTD calculated enhancement in the local electric field distribution.”.

Comment 5: The caption of Figure 2 should be extended so that the sensing approach illustrated in this figure can be better understood.

REPLY: Thanks for the suggestion. We have updated the caption of Figure 2 to make it more readable by adding the description for each step.

Figure 2. Fabrication of a miRNA sensor device integrated with antimonene

nanomaterials. Schematic illustration of the strategy employed to detect antimonene-miRNA hybridization events. □, The antimonene nanosheets was assembled on the surface of Au film. □, AuNR-ssDNA were adsorbed on the antimonene nanosheets. □, miRNA solution with different concentrations flowed through the surface of antimonene, and paired up to form a double-strand with complementary AuNR-ssDNA. □, The interaction between miRNA with AuNR-ssDNA results in release of the AuNR-ssDNA from the antimonene nanosheets. The reduction in the molecular of the AuNR-ssDNA on the SPR surface makes for a significant decrease of the SPR angle.

ACTIONS:

In the figure 2 of the page 23 in the revised manuscript, we changed the caption of Figure 2 “**Figure 2. Fabrication of a miRNA sensor device integrated with antimonene nanomaterials.** Schematic illustration of the strategy employed to detect antimonene-miRNA hybridization events. The device comprises an SPR sensor, antimonene nanomaterial, and complementary AuNR-ssDNA.” into “**Figure 2. Fabrication of a miRNA sensor device integrated with antimonene nanomaterials.** Schematic illustration of the strategy employed to detect antimonene-miRNA hybridization events. The device comprises an SPR sensor, antimonene nanomaterial, and complementary AuNR-ssDNA. The interaction between miRNA with AuNR-ssDNA results in release of the AuNR-ssDNA from the antimonene nanosheets. The reduction in the molecular of the AuNR-ssDNA on the SPR surface makes for a

significant decrease of the SPR angle.”.

Reviewers' Comments:

Reviewer #1:

Remarks to the Author:

Points:

1. The authors admit that 10 aM concentration sensitivity translates to ~30 molecules in a 5 uL sample. Although it is a significant achievement that the technique measures such low concentrations of the analyte, the measured readout is an ensemble signal from this concentration of miRNA and it is NOT a single-molecule detection. Techniques that perform detection with single-molecule sensitivity have a digitized read-out where individual molecular binding events can be identified by producing a detectable signal. Therefore this reviewer could not recommend a title with a "single-molecule" detection claim. Firstly, this will be misleading the readers. Secondly, it will be unfair to those who are working at the "single molecule" regime.

2. Authors present FDTD simulations of nanorods placed on SPR substrate with antimonene nanosheets. However, the data regarding how the additional simulations that were performed is not clear (such as illumination wavelength). Also, the authors should elaborate on the results obtained in the simulation, such as why particularly 5nm thickness gives such a significant field enhancement compared to all other thicknesses.

Reviewer #2:

Remarks to the Author:

Authors have addressed my previous concerns, so I suggest to publish the paper as is.

Reviewer #3:

Remarks to the Author:

I have carefully read the revised manuscript, the reviews provided by the other two reviewers, and the authors' response letter. I consider that the authors have fully answered all my questions and followed all my suggestions. Therefore, I recommend the publication of the paper in Nature Communications, in its current version.

Reviewer #1

Comment 1: The authors admit that 10 aM concentration sensitivity translates to ~30 molecules in a 5 μ L sample. Although it is a significant achievement that the technique measures such low concentrations of the analyte, the measured readout is an ensemble signal from this concentration of miRNA and it is NOT a single-molecule detection. Techniques that perform detection with single-molecule sensitivity have a digitized read-out where individual molecular binding events can be identified by producing a detectable signal. Therefore this reviewer could not recommend a title with a "single-molecule" detection claim. Firstly, this will be misleading the readers. Secondly, it will be unfair to those who are working at the "single molecule" regime.

REPLY: We thank the reviewer for pointing this out. We agree with the reviewer that the title should be changed. And we have used "Ultrasensitive" to replace "single-molecular" in the title. In addition, we have toned down all the claims for "single-molecular" in the manuscript.

ACTIONS:

We have changed all of "single-molecular" in the manuscript.

In the 2nd paragraph of page 5 in the revised manuscript, we have change the "Because of the extremely large adsorption energy between ssDNA and antimonene, we can envision an ultrasensitive single-molecule RNA and DNA sensor device for early cancer diagnosis." to "Because of the extremely large adsorption energy between ssDNA and antimonene, we can envision an ultrasensitive RNA and DNA sensor device for early cancer diagnosis."

In the 2nd paragraph of page 10 in the revised manuscript, we have changed the sentence "We measured the angle-resolved SPR spectra of the target miRNA-21 at the single-molecule level." to "We measured the angle-resolved SPR spectra of the target miRNA-21 at very low concentration."

In the 2nd paragraph of page 10 in the revised manuscript, we changed the sentence "The hybridization of the target miRNA-21 results in an obvious left-shift in the SPR angles even at very low concentrations of 10^{-17} M, which is at the single-molecule level." into. "The hybridization of the target miRNA-21 results in an obvious left-shift in the SPR angles even at very low concentrations of 10^{-17} M."

In the 2nd paragraph of page 11 in the revised manuscript, we changed the sentence "In particular, the LOD of miRNA sensing can approach 10 aM (~30 molecules for a 5 μ L sample), reaching the single-molecule level,^{48,49} indicating the great potential of antimonene nanosheets in applications, such as single-molecule biological imaging, clinical therapy, and environmental monitoring." into. "In particular, the LOD of miRNA sensing can approach 10 aM (~30 molecules for a 5 μ L sample), indicating the great potential of antimonene nanosheets in applications, such as single-molecule biological imaging, clinical therapy, and environmental monitoring.^{48,49}".

Comment 2: Authors present FDTD simulations of nanorods placed on SPR substrate with antimonene nanosheets. However, the data regarding how the additional simulations that were performed is not clear (such as illumination wavelength). Also, the authors should elaborate on the results obtained in the simulation, such as why particularly 5nm thickness gives such a significant field enhancement

compared to all other thicknesses.

REPLY: Thanks for the suggestion. We have added the description of FDTD in the revised manuscript.

ACTIONS:

In the 2nd paragraph of page 12 in the revised manuscript, we have added the sentences “Compared with conventional SPR using gold film, a considerable electromagnetic enhancement was observed based on the evanescent field excitation and LSPR of the AuNRs with certain thickness of antimonene nanosheets. The simulation results of the local electric field distribution around antimonene nanosheets with different thicknesses were shown in Supplementary Fig., which further verify the trend of sensitivity shown in Figure 4b. The electric field at the optimal gap (antimonene layer) between the gold film and gold nanorod is increased by approximately 300 times at incident light wavelengths of 632.8 nm, where $|E| = |E_{\text{local}}/E_{\text{in}}|$, E_{local} and E_{in} are the local and incident electric fields, respectively.”.

Reviewer #2

Authors have addressed my previous concerns, so I suggest to publish the paper as is.

Reviewer #3

I have carefully read the revised manuscript, the reviews provided by the other two reviewers, and the authors' response letter. I consider that the authors have fully answered all my questions and followed all my suggestions. Therefore, I recommend the publication of the paper in Nature Communications, in its current version.